# S²GS: Self-Supervised Gaussian Segmentation for Automatic 3D Object Scanning

## Abstract

Automatic 3D object scanning typically involves reconstructing rotating objects from images captured from different viewpoints. In such circumstances where both the object and camera are moving, existing methods need object masks for reconstruction, and the mask quality can significantly affect the final reconstruction. However, obtaining high-quality and view-consistent object masks is challenging and laborious in practice. We address this issue by introducing Self-Supervised Gaussian Segmentation (S²GS), which automatically segments the object from the background without relying on any segmentation masks. This is achieved by extending Gaussian Splatting with a learnable parameter that indicates the probability of each Gaussian belonging to the target object. We optimize this parameter using implicit object transformation constraints and regularization terms. We evaluate S²GS on our new synthetic and real datasets. Experimental results show that our approach outperforms the state-of-the-art methods (2DGS) with object masks by 27% for novel-view synthesis and 7% for geometry reconstruction.

## 1 Introduction

Accurately modeling the 3D geometry of objects remains a longstanding challenge in computer vision and graphics, yet it is essential for applications ranging from virtual reality to industrial design. Recent state-of-the-art methods, such as Neural Radiance Fields (NeRFs) (Mildenhall et al., 2021; Fu et al., 2022; Wang et al., 2021; 2023) and Gaussian Splatting (3DGS) (Kerbl et al., 2023; Guédon & Lepetit, 2024; Dai et al., 2024), have significantly advanced the field by offering unique strengths in 3D scene representation and rendering.

To achieve optimal 3D reconstruction of an object, comprehensive scans from multiple viewpoints are essential. In particular, the use of turntable setups has proven invaluable for high-throughput 3D object scanning, offering an automated, accurate, and efficient solution for various real-world applications, such as scanning cultural heritage artifacts, digital art, and sculpting. For the sake of convenience, the camera is usually fixed at discrete elevation angles rather than moving around the object. Instead, the object rotates on a turntable to allow for complete reconstruction. In this context, the capture environment often reflects casual indoor settings, where backgrounds can be complex and cluttered.

In such situations where both the camera and the object are moving, the assumption of a static scene, common in most existing methods, is violated. To address this issue, current approaches require accurate segmentation of the moving objects to mitigate the influence of the background. However, this reliance on precise segmentation limits their practical applicability, especially in scenarios where segmentation accuracy is unreliable or segmentation masks are inaccessible. To overcome this limitation, we formulate a new problem: 3D reconstruction of an object rotating on a turntable in the presence of a cluttered background without access to segmentation masks.

Figure 1 illustrates the challenges posed by background clutter in achieving accurate segmentation and 3D reconstruction using current state-of-the-art methods. In this scenario, an object is positioned on a turntable while the camera moves to capture images from various elevations using a robotic arm. Existing methods typically rely on a separate segmentation pipeline that provides object masks. This subsequently results in inaccurate 3D reconstructions as shown in the figure. Addressing this, we propose to estimate segmentation masks and 3D reconstruction simultaneously by using an additional learnable probability parameter for each Gaussian, enabling self-supervised differentiation between foreground and background.

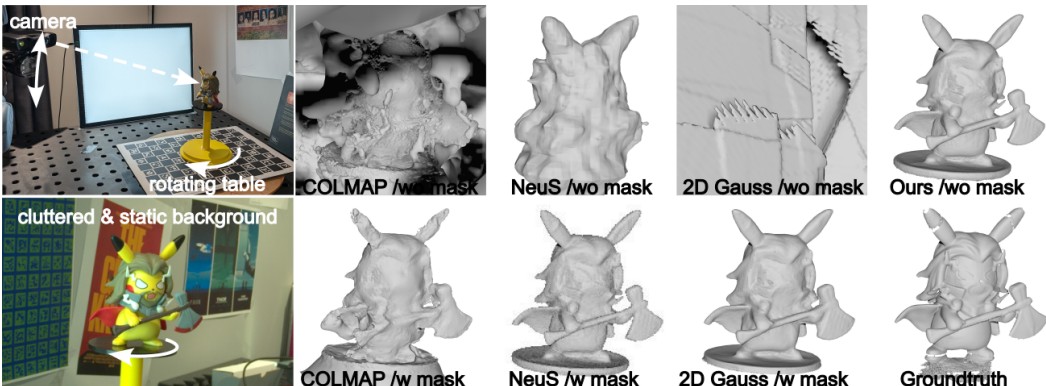

Figure 1: A cluttered background poses a serious challenge for 3D reconstruction of moving objects, for example when using a rotating table for scanning. Existing methods fail to reconstruct the object without object masks for background removal. Even when provided with object masks, these methods still perform worse than our proposed method (S²GS) without masks.

The contributions of this paper are as follows:

- Introducing the challenge of reconstructing 3D models of moving/rotating objects on turntable setups in cluttered environments without access to accurate object masks facilitates high-throughput 3D object scanning.
- Developing a novel approach for simultaneous segmentation and 3D reconstruction of moving objects in a self-supervised manner (without requiring explicit masks).
- Creating both synthetic and real datasets for the turntable scenario where both the object and camera exhibit motion. The object rotates on a turntable while the camera moves with a robotic arm.

## 2 RELATED WORK

### 2.1 NEURAL 3D RECONSTURCTION

NeRF (Mildenhall et al., 2021) can render high-fidelity novel-view images but fails to extract 3D surface from volume density field. In contrast, neural implicit functions (e.g. signed distance function (SDF) (Yariv et al., 2020) and occupancy grids (Niemeyer et al., 2020; Oechsle et al., 2021)) are usually preferred to define 3D surfaces due to accurate and smooth surface representation. Wang et al. (2021) used an SDF to represent the 3D surface and convert the signed distance value into density for rendering novel-view images via volume rendering, which enables surface reconstruction during view rendering (Yariv et al., 2021). To further enhance 3D reconstruction via novel-view synthesis, Fu et al. (2022) employs sparse 3D points, generated by Structure from Motion, to explicitly supervise the SDF network in learning 3D surfaces. Additionally, the method incorporates multi-view photometric consistency constraints to further refine and enhance the accuracy of 3D reconstruction. MonoSDF (Yu et al., 2022) leverages monocular geometric cues, such as monocular depth and normals, to refine the geometric reconstruction beyond typical rendering loss. To accelerate neural surface reconstruction, NeuS2 (Wang et al., 2023) parameterizes SDF using multi-resolution hash tables of learnable feature vectors, handling high spatial resolution with reduced computational cost. Neuralangelo (Li et al., 2023) leverages the power of multi-resolution 3D hash grids (Müller et al., 2022) and neural surface rendering to address challenges in reconstructing detailed structures from real-world scenes.This approach offers a scalable solution for high-fidelity surface reconstruction from RGB images without auxiliary data like segmentation or depth maps. By employing a progressive optimization strategy using coarse to fine hash grids, Neuralangelo captures intricate details in the reconstructed geometry. Additionally, it introduces a curvature loss to promote smooth surfaces, further enhancing the overall quality of the reconstructed geometry. Despite advancements in neural 3D surface reconstruction techniques (Wang et al., 2021; 2023; Li et al., 2023; Müller et al., 2022), challenges persist in their efficient training and rendering. These neural implicit surfaces are primarily designed for static scenes and face difficulties in adapting to dynamic scenes.

### 2.2 GAUSSIAN SPLATTING FOR 3D RECONSTRUCTION

Recently, 3DGS (Kerbl et al., 2023) has shown impressive capabilities in novel-view synthesis with real-time rendering, and has been extended to reconstruct 3D surface from multi-view images (Chen

et al., 2023; Guédon & Lepetit, 2024). By enforcing regularization terms to promote flat and well-distributed Gaussians with limited overlap, SuGaR (Guédon & Lepetit, 2024) has developed an approach that efficiently extracts accurate and editable meshes while enhancing rendering quality. 3DGSR (Lyu et al., 2024) integrates SDF with 3D Gaussians with a differentiable SDF-to-opacity transformation function, which transforms SDF values into Gaussian opacities, linking SDFs and Gaussians to enforce surface constraints and enable unified optimization. The Gaussian surfels (Dai et al., 2024) method flattens 3D Gaussian points into 2D ellipses to resolve normal ambiguity and improve surface alignment. Additionally, the method incorporates a self-supervised normal-depth consistency loss to ensure that the local z-axis aligns with the surface normal derived from rendered depth maps. 2D Gaussian Splatting (2DGS) (Huang et al., 2024) uses 2D Gaussian primitives to model and reconstruct geometrically accurate radiance fields, in which a perspective-accurate 2D splatting process utilizes ray-splat intersection and rasterization. This approach helps in accurately recovering thin surfaces and achieving stable optimization. In this work, we aim to extend the capabilities of 2DGS for 3D surface reconstruction in dynamic scenes.

## 2.3 4D DYNAMIC GAUSSIANS

The representation of 3D Gaussians has also been adapted to reconstruct the geometrical structure and appearance of a dynamic scene (Luiten et al., 2024; Yang et al., 2024; Wu et al., 2024; Yang et al., 2023b; Li et al., 2024; Liang et al., 2023; Guo et al., 2024; Wang et al., 2024; Lin et al., 2024). Dynamic 3DGS (Luiten et al., 2024) is a pioneering approach that extends 3DGS to a dynamic environment. It performs frame-by-frame optimization iteratively, effectively handling multi-view dynamic scenes with significant motion. Deformable 3DGS (Yang et al., 2024) proposes to learn a spatial-temporal deformation model (i.e. MLP) to map time-varying 3D Gaussians into a canonical space, along with a set of Gaussian in the canonical space, both of which are jointly optimized during volume rendering. Differently, 4DGS (Wu et al., 2024) modelled the deformation in a dynamic scene with multi-resolution voxel planes and a lightweight multi-head deformation decoder to further enhance the efficiency. Real-time (Yang et al., 2023b) introduces a spatial-temporal volume representation using 4D Gaussian primitives that encode both geometry and appearance. These primitives utilize parameterizations based on anisotropic ellipses and 4D spherical harmonics. To improve the modeling of dynamic scene geometry, motion-flow-based 3DGS (Guo et al., 2024; Wang et al., 2024) introduces the optical flow in the flow loss function to constraint the movement of 3G Gaussinas in 3D space. This approach allows the motion offsets of 3D Gaussians to be splatted and rendered into optical flow images. However, these dynamic 3DGS approaches focus more on modelling time-varying appearance and geometry and fail to reconstruct accurate 3D structures.

## 3 PRELIMINARY: 2D GAUSSIAN SPLATTING

Since we focus on achieving high-quality reconstruction, our method builds upon the state-of-the-art surfel-based 2DGS (Huang et al., 2024) due to its superior geometry performance and efficiency. 2DGS proposes to collapse the 3D volume into a set of 2D-oriented planar Gaussian disks and introduces a perspective-accurate 2D splatting process. Similar to 3DGS, the 2D splat is characterized by its central point $p_k$, two principal tangential vectors $t_u$ and $t_v$, and a corresponding scaling vector $S = (s_u, s_v)$ that controls the variances of the 2D Gaussian distribution. Compared to 3DGS, 2DGS represents the geometry of the scene better because the oriented planar Gaussian can be perfectly aligned to the surface and the normal direction is well-defined as the normal of the plane.

To be specific, a 2D Gaussian disk is defined in a local tangent space in world space, parameterized as:

$$P(u, v) = \mathbf{x} + s_u \mathbf{r}_u u + s_v \mathbf{r}_v v \tag{1}$$

And for the point $\mathbf{u}(u, v)$ in the $uv$ space, its 2D Gaussian value can then be calculated by standard Gaussian

$$\mathcal{G}(\mathbf{u}) = \exp\left(-\frac{u^2 + v^2}{2}\right) \tag{2}$$

The center $\mathbf{x}$, scaling $(s_u, s_v)$, and the rotation $(\mathbf{r}_u, \mathbf{r}_v)$ are learnable parameters. Following 3DGS (Kerbl et al., 2023), each 2D Gaussian primitive has opacity $\alpha$ and view-dependent appearance $c$ parameterized with spherical harmonics.

Instead of projecting the 2D Gaussian primitives onto the image space for rendering, 2DGS derives the intersection point of ray and splat in the local tangent space by plane intersection, which alleviates the problem of splat degeneration, especially at grazing angle. The rasterization process is

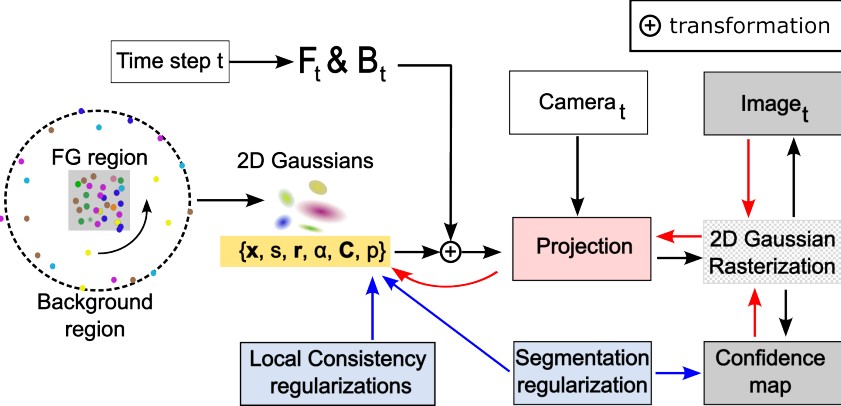

Figure 2: Overiew of our proposed S2GS framework which improves 3D reconstruction of moving rigid objects without the need for input masks. $\{\mathbf{x}, \mathbf{s}, \mathbf{r}, \alpha, \mathbf{C}, p\}$ are learnable and represent the 3D location of a 2D Gaussian, its 2D scale, 3D rotation, transparency, spherical harmonics, and probability as foreground. $F_t$ and $B_t$ are the transformations at frame $t$ from the local to the world coordinates for the dynamic foreground and static background respectively. Black arrows are operation flows, red arrows gradient flows, and blue arrows regularization flows.

similar to 3DGS, in that 2D Gaussians are sorted based on the depth of their center and volumetric alpha blending is used to integrate alpha-weighted appearance from front to back:

$$\mathbf{c}(\mathbf{x}) = \sum_{i=1} \mathbf{c}_i \alpha_i \hat{G}_i(\mathbf{u}(\mathbf{x})) \prod_{j=1}^{i-1} (1 - \alpha_j \hat{G}_j(\mathbf{u}(\mathbf{x}))) \qquad (3)$$

## 4 METHOD

In this section, we introduce our new approach for 3D reconstruction of moving objects, particularly in environments with cluttered backgrounds. Our method eliminates the need for explicit object masks by employing a fully self-supervised Gaussian segmentation technique.

### 4.1 PROBLEM FORMULATION

Cluttered backgrounds pose significant challenges for 3D reconstruction of objects on rotating tables. Existing methods often struggle to reconstruct objects without explicit background masks. While some methods perform better when provided with masks, they are still affected by the mask quality, which is especially for low for a cluttered background and a complex foreground object. So there is a need for a better approach that does not depend on input masks for background-foreground separation.

As shown in Fig. 1, our goal is to address the challenge of reconstructing 3D models from multiple views in scenarios involving moving objects. For demonstration, this turntable setup is typical in 3D object scanning setups where the object rotates on a rotating table, and cameras capture images from discrete elevation angles. Often in production environments, unlike a lab environment, the scene includes cluttered, static backgrounds that are difficult to remove, posing challenges for accurately reconstructing the foreground object. Errors in background removal can severely impact the accuracy of 3D reconstruction.

To be specific, our approach involves partitioning the scene into two distinct regions: the dynamic foreground and the static background. The foreground region encompasses the target object that rotates with the turntable. Unlike scenarios involving dynamic scenes (Pumarola et al., 2021; Luiten et al., 2023; Yang et al., 2024), we assume the foreground object is rigid, and its movement is consistent across the region. The background region encompasses all other static elements in the scene.

Given a set of images $\mathcal{I} = \{I_t \mid t = 1, 2, \ldots, n\}$ captured at $n$ discrete time steps, several transformations are associated with the camera and the scene. Firstly, the camera poses $\mathcal{T} = \{T_t \in \mathbb{R}^{4 \times 4} \mid t = 1, 2, \ldots, n\}$ defined as the transformation from the camera to the world

coordinate system by:

$$X_t = T_t \cdot x^c \tag{4}$$

where $x^c$ is the homogeneous coordinates in the camera coordinate system and $X_t$ is the corresponding coordinate in the world coordinate system at time step $t$.

Secondly, we define the transformations $\mathcal{F} = \{F_t \in \mathbb{R}^{4 \times 4} \mid t = 1, 2, \ldots, n\}$ from the dynamic foreground to the world coordinate system by:

$$X_t = F_t \cdot x^f \tag{5}$$

where $x^f$ is the homogeneous coordinate in the foreground coordinate system. Similarly, we also define the transformations $\mathcal{B} = \{B_t \in \mathbb{R}^{4 \times 4} \mid t = 1, 2, \ldots, n\}$ from the static background to the world coordinate system by:

$$X_t = B_t \cdot x^b \tag{6}$$

where $x^b$ is the homogeneous coordinate in the background coordinate system.

Note that the local coordinates $x^f$ and $x^b$ are irrelevant to the time step $t$ because we assume the scene is rigid without any deformation. Since the background is always static, $B_t$ is actually all the same across different time stamps. Furthermore, by setting $\{B_t = I \mid t = 1, 2, \ldots, n\}$ where $I \in \mathbb{R}^{4 \times 4}$ is the identity matrix, we can simplify the representation by aligning the world coordinate system rightly with the background coordinate system without loss of generality. Thus, we have the relation $X_t^w = x^b$ for any point in the background region at any time.

We note that these poses can be estimated by running a structure-from-motion method (Schönberger & Frahm, 2016) on the two separated regions or pre-calibration of the camera set-up and thus are assumed to be known in this paper. The proposed situation is common in practice but poses challenges for existing methods due to issues with camera projection transformations. Most methods assume that the foreground object is static relative to the background, which is typically true in controlled environments. However, in the context of scanning with a rotating turntable, this assumption breaks down because both foreground and background are captured by the same camera but have different motion characteristics. If we apply the camera transformation intended for the foreground object to the background, or vice versa, it would result in inconsistencies. If this inconsistency is not properly managed, it can lead to noticeable artifacts in the reconstructed results.

The most straightforward approach to this challenge is image matting. While SAM (Kirillov et al., 2023) and its variants (Zou et al., 2024; Ke et al., 2024) deliver remarkable results in image segmentation, they still require human intervention, as the user must manually prompt the target object to achieve high-quality matting in practical applications. This extra manual step complicates the reconstruction pipeline, making it difficult to fully automate the process, which in turn reduces its scalability for high-throughput 3D scanning. Moreover, despite the advances in large-scale models, obtaining accurate pixel-level image masks across all viewpoints remains a significant hurdle.

### 4.2 EXTENDED 2D GAUSSIAN

As analyzed in Section Sec. 4.1, a crucial aspect of addressing this problem involves accurately separating the static and dynamic regions to process each area appropriately. In this section, we outline our approach to achieving segmentation in a fully self-supervised manner. Firstly, we adopt the 2D Gaussians from (Huang et al., 2024) as our scene representation method. This choice is motivated by the method's use of point-based representation, which explicitly defines the scene using thousands of Gaussian primitives. This approach allows us to easily assign customized attributes to the Gaussians. Compared to the neural implicit representations, such as neural radiance field (Mildenhall et al., 2021; Chen et al., 2022) and neural SDF (Wang et al., 2021; Oechsle et al., 2021; Wang et al., 2023), the explicit representations are more suitable for addressing the formulated problem.

As in the original 2DGS (Huang et al., 2024), the scene is represented by 2D Gaussian primitives which comprise of a series of learnable parameters $\{\mathbf{x}_i, \mathbf{s}_i, \mathbf{r}_i, \alpha_i, \mathcal{C}_i\}$, where $\mathbf{x}_i \in \mathbb{R}^3$ denotes the position of the Gaussian's center, $\mathbf{s}_i \in \mathbb{R}^2$ is the scale of the 2D Gaussian, $\mathbf{r}_i \in \mathbb{R}^4$ is its rotation (orientation) represented by a quaternion, $\alpha_i \in \mathbb{R}$ is the opacity, and $\mathcal{C}_i \in \mathbb{R}^k$ is spherical harmonic coefficients of the Gaussian primitive. Apart from the standard parameters, we aim to assign an additional indicator parameter that identifies whether each Gaussian is static or dynamic. This distinction allows us to apply appropriate transformations to the static and dynamic Gaussians accordingly. In case that a native binary indicator is not differentiable and is not capable to be

optimized by gradient descent. We propose to use a continuous parameter $p \in [0, 1]$ to indicate the probability of a Gaussian primitive belonging to the foreground region. Ideally, for any dynamic Gaussian in the foreground should have $p = 1$, while others in the background will have $p = 0$. Thus, the learnable parameters of a 2D Gaussian primitive are extended to be $\{\mathbf{x}_i, \mathbf{s}_i, \mathbf{r}_i, \alpha_i, \mathcal{C}_i, p_i\}$.

In the next section, we will talk about how we optimize the probability $p$ together with other parameters in a self-supervised manor.

### 4.3 Self-supervised Gaussian Segmentation

We realize fully self-supervised Gaussian segmentation by integrating the foreground probability $p$ into the rendering process differentially.

As described in Sec. 4.1, we segment the scene into foreground and background regions and each region is related to the world coordinate system by the transformations $\mathcal{F}$ and $\mathcal{B}$. Instead of putting everything in the world coordinate system directly like other Gaussian Splatting methods (Kerbl et al., 2023; Huang et al., 2024), we initialize all the Gaussians in the local coordinate systems. In the rendering process, we transform the Gaussians to the world coordinate system by the transformations $\mathcal{F}$ and $\mathcal{B}$.

The transformation is performed probabilistically according to the foreground probability $p_i$. To be specific, given a Gaussian primitive whose center position is $\mathbf{x}_i$ in its local coordinate system, it will be transformed to the world coordinate system at time step $t$ by:

$$X_i^t = p_i \mathrm{F}_t \cdot \mathbf{x}_i + (1 - p_i) \mathrm{B}_t \cdot \mathbf{x}_i \tag{7}$$

where $X_i^t$ is the coordinate of the transformed center point in the world coordinate system at time step $t$. Intuitively understanding of the formulation, we calculate the expectation of the transformed Gaussian's center according to the foreground probability $p_i$. With the proposed soft version of the aforementioned transformation Sec. 4.1, the process is fully differentiable, and the probability parameter $p_i$ can be optimized with the photometric loss by gradient descent. We don't use any explicit supervision for the probability $p_i$.

As the training process is converging, the probability $p_i$ is expected to be pushed closely to either $0$ or $1$ and Eq. (7) will degenerate to either Eq. (5) or Eq. (6) meaning the Gaussian primitive is correctly transformed to the world coordinate system. In this way, we can segment the Gaussians into foreground and background automatically.

Recall the parameters of our 2D Gaussian $\{\mathbf{x_i}, \mathbf{s_i}, \mathbf{r_i}, \alpha_i, \mathcal{C}_i, p_i\}$, apart from the center position $\mathbf{x}_i$ which we have already transformed, we also transform the rotation-related parameters rotation vector $\mathbf{r}_i$ and spherical coefficients $\mathcal{C}_i$ from the local space to the world space. We don't use the same soft transformation for the rotation vector and spherical coefficients because both $\mathbf{r}_i$ and $\mathcal{C}_i$ are only related to the $SO3$ rotation transformation. According to Hartley et al. (2013), linear interpolation in $SO3$ space is not straightforward, and it will add complexity to the optimization problem. Therefore, we choose to transform the left parameters in a 'hard' way. Firstly, we decide the attribute of the Gaussian by a thresholding strategy. For each Gaussian, the binary indicator $P_i$ is achieved by:

$$P_i = \begin{cases} \text{True} & \text{if } p_i \geq \tau \\ \text{False} & \text{if } p_i < \tau \end{cases} \tag{8}$$

where $\tau$ is a hyperparameter and we empirically set $\tau = 0.8$ for all the experiments.

Then, we transform the rotation vector $\mathbf{r}_i$ and spherical coefficients $\mathcal{C}_i$ according to the corresponding transformations $\mathcal{F}$ or $\mathcal{B}$.

For a rotation vector $\mathbf{r}_i$ of a foreground Gaussian, we apply the rotation component of the transformation $\mathcal{F}$ directly by:

$$\mathbf{r}_i^t = \mathcal{R}^{-1}(\mathrm{F}_t^{3 \times 3} \cdot \mathcal{R}(\mathbf{r}_i)) \tag{9}$$

where $\mathcal{R}$ is the mapping from a quaternion to a rotation matrix, and $\mathrm{F}_t^{3 \times 3}$ is the rotation component of the transformation $\mathrm{F}_t$ and $\mathbf{r}_i^t$ is the transformed rotation vector at time step $t$. For the sake of simplicity, we omit the transformation for background Gaussians as it's the same as that for the foreground and it's usually assumed to be identical.

We also transform the spherical coefficients $\mathcal{C}_i$ with the help of Wigner D-matrix (Wigner, 1931). Please refer to the supplementary material for more details.

## 4.4 Optimization

In this section, we will introduce the loss functions we proposed to facilitate training, especially for the self-supervised Gaussian segmentation. Since we have no idea of where the background or the foreground is, we use three regularization loss functions.

**Local Consistency Loss**. As a prior knowledge, we know that the Gaussians belonging to the same region tend to gather while those from different regions are usually away from each other. That is to say, the distribution of foreground probability $p$ should be consistent within a local region. Based on the analysis, we propose a Local Consistency Loss similar to (Ye et al., 2023) to regularize the learning of the probability $p_i$. For each Gaussian, the loss encourages other $k$-nearest Gaussians to have a similar distribution of the foreground probability. As shown in Eq. (10), for each Gaussian, we define the $\mathcal{L}_{lc}$ as the KL divergence of the foreground probability.

$$\mathcal{L}_{lc} = \frac{1}{k} \sum_{j=1}^{k} D_{KL}(p_i || p_j) \tag{10}$$

**3D segmentation regularization.** As we discussed in Sec. 4.2, the parameter $p_i$ is defined as the probability that a Gaussian belongs to the foreground. With the proceeding of the training process, $p$ should be more deterministic that it becomes close to either $0$ or $1$. We use the point-wise loss $\mathcal{L}_{3ds}$ to force the probability of each Gaussian to be deterministic.

$$\mathcal{L}_{3ds} = \frac{1}{k} \sum_{i=1}^{k} -(p_i \log(p_i) + (1 - p_i) \log(1 - p_i)) \tag{11}$$

**2D segmentation regularization.** We also apply similar regularization in the 2D image space. A confidence map $\mathcal{C}$ is rendered by replacing the color information $\mathbf{c}_i$ with the probability $p_i$ in Eq. (3). Then, a similar regularization $\mathcal{L}_{2ds}$ is also applied to the confidence map by:

$$\mathcal{L}_{2ds} = \mathbb{E}[-(\mathcal{C} \log(\mathcal{C}) + (1 - \mathcal{C}) \log(1 - \mathcal{C}))] \tag{12}$$

**Overall.** We also incorporated the losses proposed in the original 2DGS (Huang et al., 2024) paper $\mathcal{L}_{ori}$ including the normal consistency loss and photometric loss. Combined them together, the total loss $\mathcal{L}$ used for training is

$$\mathcal{L} = \mathcal{L}_{ori} + \lambda_{lc}\mathcal{L}_{lc} + \lambda_{3ds}\mathcal{L}_{3ds} + \lambda_{2ds}\mathcal{L}_{2ds} \tag{13}$$

## 4.5 Implementation details

**Initialization.** To minimize manual labor, we did not use SfM points to initialize the point cloud. Instead, we adopted a random initialization approach. As described earlier, our Gaussians are defined in the local coordinate systems. For the foreground region, we randomly initialize Gaussians within a cube region centered at the origin. As for the background region, we simply initialize Gaussians roughly on a sphere with 4 times the radius than the side length of the cube with probability. The foreground probability is initialized to be $p_i = 0.9$ for Gaussians in the foreground region and $p_i = 0.1$ in the background region. we randomly initialize other parameters.

**Optimization** Our work is implemented by PyTorch and CUDA and optimized by the Adam optimizer. All the experiments are conducted on a single NVIDIA RTX4090 GPU. We follow the training process of 2DGS that we train 30000 iterations in total. For the training loss, we assign $\lambda_{lc} = 50$, $\lambda_{3ds} = 0.1$ and $\lambda_{2ds} = 0.1$ for the real dataset and $\lambda_{lc} = 50$, $\lambda_{3ds} = 0.01$ and $\lambda_{2ds} = 0.01$ for the synthetic dataset, respectively. The regularization losses $\mathcal{L}_{lc}$, $\mathcal{L}_{3ds}$ and $\mathcal{L}_{2ds}$ are enabled after 5000 iterations.

## 5 Experiments

### 5.1 Experimental Settings

**Datasets** We captured a real dataset consisting of 9 objects using a motorized rotating table and a Zivid-Two [1] camera mounted on a UR5e [2] robot arm as shown in the top left inset of Figure 1. The

---

[1] https://www.zivid.com/zivid-2
[2] https://www.universal-robots.com/products/ur5-robot/

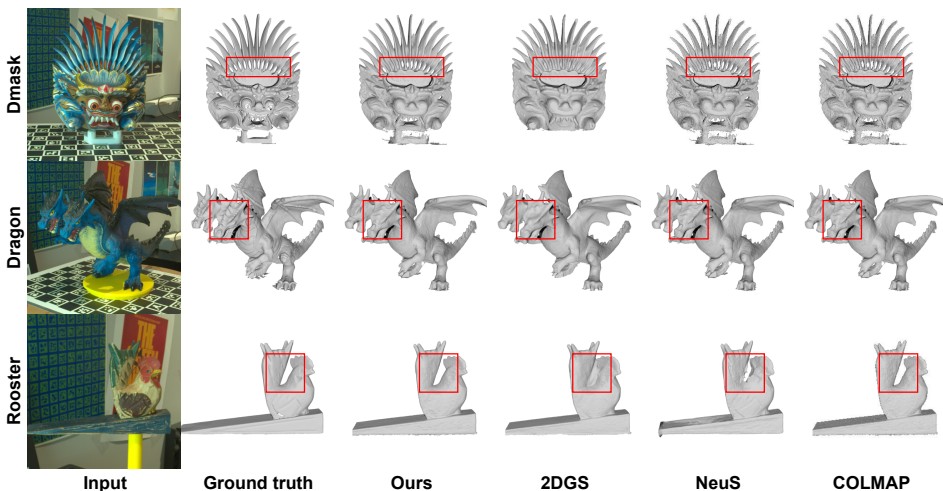

Figure 3: Qualitative reconstruction results on the real dataset.

camera captured RGB-D data at different rotation steps and tilting angles. The depths are converted to point clouds and fused to create a ground truth mesh for comparison.

We also create a synthetic dataset consisting of 9 objects in 3 carefully designed scenarios. The dataset is rendered with Blender (Community, 2018) Cycles engine with a resolution of $800 \times 800$. The camera is set at 5 different elevation angles and the foreground object will rotate around for each camera position. Please refer to the Appendix B for more information of the datasets.

**Baselines**  In our experimental evaluation, we select four baseline methods to compare with. COLMAP (Schönberger et al., 2016), a conventional Structure from Motion method that reconstructs 3D structures from 2D images using feature matching and optimization. NeuS (Wang et al., 2021) combines and integrates SDF into the NeRF framework for high-quality surface reconstruction and realistic novel-view synthesis. 2DGS (Huang et al., 2024) uses 2D Gaussian primitives to model and reconstruct geometrically accurate radiance fields, enhancing surface alignment and real-time rendering. Lastly, Deformable 3DGS (D-3DGS) (Yang et al., 2024) extends 3D Gaussian Splatting to dynamic scenes by modeling changes in geometry and appearance over time, making it ideal for applications involving motion and varying conditions. These baselines provide a comprehensive comparison for our proposed method.

For the first three baseline methods, we perform two sets of experiments: with object masks and without. We use segment and tracking anything (Kirillov et al., 2023; Ke et al., 2024; Yang et al., 2023a) to generate object masks. Examples of object masks are shown in Fig. 5.

| | Methods | Bear | Captain | Controller | Dmask | Dog | Dragon | Pikachu | Plant | Rooster | Avg. |
|---|---|---|---|---|---|---|---|---|---|---|---|
| w/ mask | COLMAP | 0.0663 | 0.2442 | 0.2246 | 0.2089 | 0.1656 | 4.2967 | 0.4280 | 16.7652 | 0.9057 | 2.5895 |
| | NeuS | 0.0695 | 0.1082 | 0.1484 | 0.1804 | 0.0964 | 0.1247 | 0.2990 | 0.1707 | 0.1839 | 0.1535 |
| | 2DGS | 0.0690 | 0.1153 | 0.1815 | 0.1366 | 0.1063 | 0.1054 | 0.2514 | 0.1458 | 0.1170 | 0.1365 |
| w/o mask | COLMAP | 34.6392 | 45.4321 | 44.8083 | 35.4063 | 35.6062 | 32.4344 | 35.1575 | 41.1090 | 33.0436 | 37.5152 |
| | NeuS | 1.5605 | - | 1.4726 | 2.4949 | - | 2.4661 | 2.5834 | 2.2161 | - | 2.1323 |
| | 2DGS | 26.4271 | 12.4420 | 23.9106 | 18.7352 | - | 20.5928 | 26.6814 | 24.6871 | 26.1505 | 22.4533 |
| | D-3DGS | 2.1397 | 1.6192 | 1.9330 | 1.5135 | 0.5591 | 6.3222 | 0.7764 | 6.1672 | 1.8238 | 2.5393 |
| | S$^2$GS (ours) | 0.0769 | 0.1210 | 0.1166 | 0.1075 | 0.0860 | 0.1044 | 0.2966 | 0.1249 | 0.1111 | 0.1272 |

Table 1: The Chamfer-$\mathcal{L}_1 \downarrow$ distance of 3D reconstruction results on the real dataset. We color each cell as best , second , and third . "−" indicates the method failed to reconstruct a mesh. All results are scaled up by $100\times$ for better comparison.

## 5.2 EXPERIMENTAL RESULTS

### 5.2.1 3D SURFACE RECONSTRUCTION.

Table 1 and Table A1 present the reconstruction quality measured by the Chamfer-$\mathcal{L}_1$ distance. S$^2$GS outperforms the other baselines, achieving the best average Chamfer distance in both settings (**w/** and **w/o** mask). It consistently ranks first in most categories, indicating its robustness and accuracy in diverse conditions. Notably, our method even slightly outperforms the base method 2DGS in the

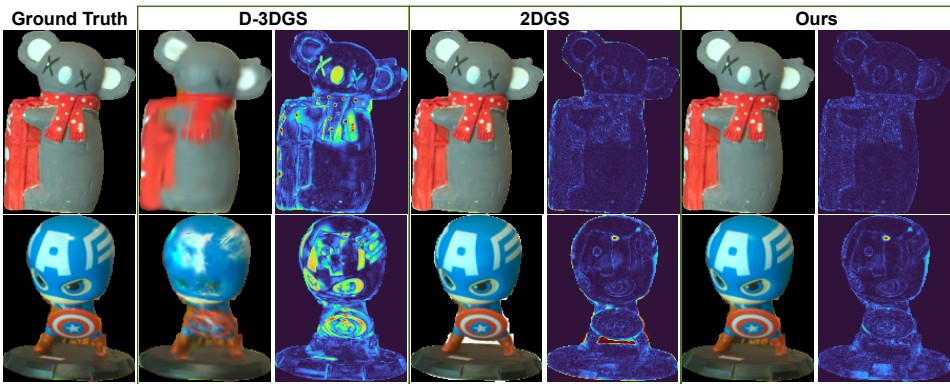

Figure 4: Novel-view synthesis and corresponding error maps (MSE) on the real dataset.

masked scenario. We attribute this improvement to the fact that other methods rely on object masks for reconstruction, and imperfect masks can negatively affect their performance. In contrast, our self-supervised method accurately segments the object, especially in challenging regions.

As shown in Fig. 3, our approach produces reconstructions with fine details and accurate geometry. When reconstructing sharp, outward spikes in the *Dmask*, baseline methods fail to capture these features accurately: 2DGS tends to produce exaggerated edge effects, while NeuS exhibits eroded features. In contrast, our method effectively captures these intricate details, significantly outperforming the baselines. Similarly, in the case of the *Rooster*, the adverse impact of noisy masks on 2DGS and NeuS is evident, particularly in the gap between the head and tail. 2DGS erroneously reconstructs this gap as part of the foreground object, leading to inaccuracies. However, our method effectively segments the object and background components, maintaining a clear distinction and avoiding such errors. This underscores the robustness of our approach in reconstructing objects without reliance on masks.

|  | Methods | Real dataset | | | Synthetic dataset | | |
|---|---|---|---|---|---|---|---|
|  |  | PSNR ↑ | SSIM ↑ | LPIPS ↓ | PSNR ↑ | SSIM ↑ | LPIPS ↓ |
| w/ mask | NeuS | 21.35 | 0.854 | 0.149 | 26.96 | 0.902 | 0.058 |
|  | 2DGS | 25.10 | 0.915 | 0.065 | 26.16 | 0.931 | 0.063 |
| w/o mask | NeuS | 18.36 | 0.456 | 0.458 | 19.41 | 0.648 | 0.280 |
|  | 2DGS | 12.00 | 0.469 | 0.637 | 15.64 | 0.693 | 0.471 |
|  | D-3DGS | 23.42 | 0.807 | 0.201 | 19.76 | 0.787 | 0.201 |
|  | S2GS (ours) | 31.86 | 0.946 | 0.047 | 33.89 | 0.969 | 0.037 |

Table 2: Novel-view synthesis metrics on both real and synthetic datasets. We report the PSNR ↑, SSIM ↑, and LPIPS ↓ for rendering quality.

### 5.2.2 NOVEL-VIEW SYNTHESIS.

Table 2 presents a detailed comparison for the novel-view synthesis task. Results show that S²GS outperforms others by a large margin. We provide novel-view renderings and visualize the corresponding mean squared error (MSE) maps in Fig. 4. Compared to D-3DGS without masks, our method achieves a substantial improvement in rendering quality. When comparing with NeuS and 2DGS under the masked condition, our method generally shows an improvement of over $20\%$. From the error map comparison with 2DGS, we observe that S²GS maintains performance comparable to 2DGS in the interior region, which is expected, but exhibits significantly less error at the boundaries due to correctly modeling the background.

### 5.2.3 ANALYSIS OF THE OBJECT SEGMENTATION MASK.

We evaluate the segmentation mask quality on the synthetic dataset by comparing the results of 2DGS, SAM, and our proposed method, S²GS. For 2DGS, segmentation masks are generated based on opacity, while for S²GS, we leverage the confidence map. The results indicate that our method achieves superior segmentation accuracy compared to 2DGS and even outperforms SAM, as measured by mask IoU. This demonstrates that our self-supervised method effectively handles challenging segmentation scenarios, improving overall reconstruction quality.

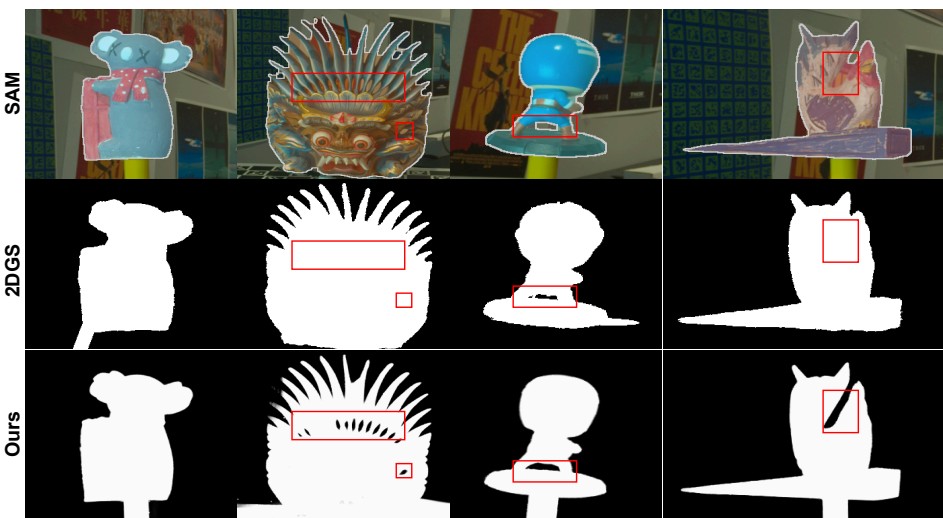

Figure 5: Qualitative analysis of the segmentation masks.

The segmentation mask of the real dataset is shown in Fig. 5. As shown in the first row, although SAM generally achieves compelling segmentation results, it struggles to segment perfectly in difficult regions (see the *Dmask* and *Rooster*). Errors in the segmentation masks are propagated to 2DGS, leading to incorrect segmentation. Similar to the reconstruction results in Fig. 3, our method achieves precise segmentation even in challenging regions. This analysis further demonstrates the value of our approach, showing that we can improve reconstruction efficiency while also overcoming segmentation mask inaccuracies.

### 5.2.4 ABLATION STUDY.

Table 4 demonstrates the impact of different regularization components on the base model, evaluated on the real dataset using Chamfer-$\mathcal{L}_1$ distance and PSNR metrics. Adding regularization terms to the base model results in improvements in both metrics. However, the full model, which incorporates all three regularization terms, achieves the best Chamfer-$\mathcal{L}_1$ but exhibits a slight decrease in PSNR, indicating trade-offs associated with combining all regularization terms. Overall, these results highlight that incorporating specific regularization terms can enhance the geometric reconstruction quality of the 3D model. The local consistency loss term $\mathcal{L}_{lc}$ proves to be highly effective in improving both metrics. The segmentation regularization terms ($\mathcal{L}_{2ds}$ and $\mathcal{L}_{3ds}$), which aid in separating moving objects from the static background, enhance geometric reconstruction by achieving the best geometric accuracy but at the cost of a slight decrease in rendering quality.

| Method | 2DGS | SAM | **Ours** |
|---|---|---|---|
| IoU | 0.876 | 0.933 | **0.977** |

Table 3: Analysis of the mask quality. We report the mask IoU on the synthetic dataset.

| Metrics | base | $+\mathcal{L}_{lc}$ | $+\mathcal{L}_s + \mathcal{L}_{lc}$ | full |
|---|---|---|---|---|
| Chamfer-$\mathcal{L}_1$ ↓ | 0.1328 | 0.1296 | 0.1302 | 0.1272 |
| PSNR ↑ | 32.25 | 32.85 | 32.93 | 31.86 |

Table 4: Ablation study of regularization terms on the real dataset. We report Chamber-$\mathcal{L}_1$ for 3D reconstruction and PSNR for novel-view synthesis.

## 6 CONCLUSION

We propose a novel method, S$^2$GS, for self-supervised segmentation of Gaussian splats in the 3D reconstruction of moving objects, particularly applicable to automatic 3D scanning. Our approach demonstrates robustness to cluttered backgrounds, enabling high-quality 3D model reconstruction without the need for object masks. Compared to other state-of-the-art methods, although Deformable 3D Gaussian Splatting (Yang et al., 2024) operates without object masks, it performs significantly worse than our method. Furthermore, we outperform our base method 2DGS even when it utilizes high-quality object masks. Future work involves extending the self-supervised Gaussian segmentation feature of S$^2$GS to more general moving object scenarios.

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
