# OpenReview forum: "S$^2$GS: Self-supervised Gaussian Segmentation for Automatic 3D Object Scanning"
_ICLR.cc/2025/Conference — ICLR 2025 Conference Withdrawn Submission_

### Official Review · Reviewer_9gZX · 2024-10-29

**Soundness:** 2
**Presentation:** 3
**Contribution:** 2
**Rating:** 3
**Confidence:** 4

**Summary:**

This paper introduces a novel approach for the automatic segmentation of foreground dynamic objects. In a turntable setting, where the foreground object rotates over time against a static background, the proposed method leverages 2D Gaussian Splatting as its foundational pipeline and incorporates an additional indicator parameter to distinguish whether a Gaussian group is static or dynamic. During training, dynamic Gaussian groups are translated through foreground transformation and are used to render the final images in conjunction with static Gaussian groups. This technique enables the automatic segmentation of the rotated foreground object.

**Strengths:**

1. The method is self-supervised and does not rely on any masks for supervising the segmentation process.
2. It collects a valuable dataset for related tasks.

**Weaknesses:**

My major concern is that the data is very simple and the method is hard to generalize. All experiments are conducted in a turntable setup, where the moving objects rotate along a single axis without any positional translation. The shapes and poses of the objects remain unchanged, which is not very common in real-world cases. Although the authors assert that they assume no deformation in the foreground and background, this setup is significantly different  from real-world scenarios. Consequently, this method might be only applicable in very limited cases. The occurance of any deformation or complex translation in the object could result in segmentation failure.

**Questions:**

1. Could the authors provide more details on how \( F_t \) is calculated? It appears that the authors claim it can be obtained through Structure-from-Motion (SfM) in line 236, and they directly use ground truth values. However, the subsequent sentences discuss the limitations of existing methods for pose estimation. Determining the pose of dynamic rigid objects is not straightforward, yet the proposed method does not address these pose estimation challenges. I hope the authors can expand on the pose estimation process or discuss its potential impact on the final results.

2. The authors assume that the pose of the foreground object is known, which is an uncommon assumption from my perspective. While this could render the method purely self-supervised, it may not be very practical in real-world applications. If some tracking and segmentation methods like [1], [2] are introduced, I think that no additional training is required, and the dynamic objects can also be segmented successfully. It might sightly influence the metrics according to the paper, but it can be applied to more real-world examples. I hope that the authors can do some experiments on this.

[1] Wang, Qianqian, et al. "Tracking everything everywhere all at once." Proceedings of the IEEE/CVF International Conference on Computer Vision. 2023.
[2] Rajič, Frano, et al. "Segment anything meets point tracking." arXiv preprint arXiv:2307.01197 (2023).

---

### Official Review · Reviewer_LdFk · 2024-11-01

**Soundness:** 2
**Presentation:** 1
**Contribution:** 2
**Rating:** 5
**Confidence:** 4

**Summary:**

This paper aims to address the challenge of reconstructing 3D models of moving or rotating objects on turntable setups in cluttered environments without the use of explicit object masks.

* This paper proposes pre-calculating the transformation matrices for the foreground object and camera pose using Structure from Motion (SFM), and then conducting the optimization of segmentation and 3D reconstruction.
* This paper conducts experiments on two created datasets, including both a real dataset and a synthetic dataset. The results demonstrate that the proposed method achieves improved performance.

**Strengths:**

* The proposed method introduces a new solution for surface reconstruction in settings involving moving object without explicit masks. The key idea of the method is to explicitly optimize the probability that each Gaussian belongs to the foreground object.
* The paper creates both synthetic and real datasets and conducts experiments that demonstrate the proposed method achieves improved performance compared to baseline methods.

**Weaknesses:**

* The method's setting may not be practical. Specifically, the proposed method assumes that the transformation matrices for the foreground objects and camera poses can be pre-computed using an existing SFM method (e.g., COLMAP). However, obtaining these transformation matrices for the foreground objects and camera poses is the key challenge in the 3D reconstruction of moving objects, as shown in Figure 1 (COLMAP). Could the author clarify how they envision obtaining these transformation matrices in real-world scenarios without masks? Additionally, it is necessary to discuss potential limitations or assumptions in their approach regarding the availability of these matrices. Lastly, it is strongly recommended to compare the difficulty of obtaining these matrices versus obtaining explicit masks in practical applications.

* The paper lacks important details. Specifically, the meaning of the term "poses" mentioned in line 236 is unclear. Additionally, the paper does not provide a detailed procedure for how the "poses" used were obtained. Moreover, the mention of "two separated regions" in line 235 raises the question of whether the preprocessing still requires mask information. Could the author clarify these points?

* The paper's writing needs improvement, particularly in the methodology description. For example, the sentence in line 198, "which is especially for low for a cluttered background and a complex foreground object," is confusing. The paragraph from lines 312 to 317 appears unorganized. I strongly suggest thorough proofreading to enhance the overall readability.

**Questions:**

Besides the following question, please refer to the weakness section above for my concerns.

* The author states in line 262, "the explicit representations are more suitable for addressing the formulated problem." Could the author provide more explanation, as the conclusion is not obvious?

---

### Official Review · Reviewer_jHbf · 2024-11-03

**Soundness:** 2
**Presentation:** 3
**Contribution:** 2
**Rating:** 6
**Confidence:** 3

**Summary:**

This paper proposes an object segmentation method when reconstructing the object by 2DGS. The method is simple, which adds an additional object indicating parameter into the parameter list of 2DGS. To account for foreground and background, the method involves many additional transformations between coordinate systems. The method is able to reconstruct moving objects captured by moving cameras.

**Strengths:**

1. Adapt 2DGS to the object segmentation task
2. Optimization with various regularization methods.

**Weaknesses:**

1. The results are not from outdoor?
2. No background about object segmentation by 3DGS.

**Questions:**

1. In order to reconstruct an object, we need to either rotate the object or moving the camera. Why do you rotate the object and moving the camera both?

2. How the cluttered background influences your result? Do you conduct ablation study on the degree of clutter in the background? Some examples in the paper have very complex background, for which you can still reconstruct perfect meshes. Can the method always output perfect results? Please give more explanations.

3. Could you please visualize the learned indicating parameters in the 3D space?

4. How much time do you need? Is there advantage compared with 2DGS with mask?

5. The paper is well-written, but there is no related work about object segmentation in the framework of 2DGS/3DGS. As far as I know, there are some works studying 3DGS-based object segmentation.

6. In figure 5, could you please interpret more about why your method can recognize holes of the object.

---

### Official Review · Reviewer_vQpz · 2024-11-03

**Soundness:** 2
**Presentation:** 2
**Contribution:** 2
**Rating:** 5
**Confidence:** 3

**Summary:**

This submission considers the problem of 3D reconstruction from automatic object scanning process. The key contribution is a Self-Supervised Gaussian Segmentation module, which is constructed by formulating per-pixel foreground/background label as a learnable parameter. The proposed method gets rid of dependency on external segmentation tools, or initialization from SfM methods.

**Strengths:**

+ This work proposes a direct treatment on foreground/background segmentation in Gaussian splatting framework, which is more integrated than the prior works depending on external cues such as SAM.

+ The proposed method can start with a randomized initialization, which further get rid of dependency on external SfM methods.

**Weaknesses:**

-	My main concern lies on the practical utility of the proposed setting. The real-world data collection procedure (which obtains the images in the accompanied video) does not really justify the necessity of background removal – it seems easier and plausible to set a green screen as 1) the camera moves within a small vertical range; 2) the objects of interest are all of small dimension.
-	The running time efficiency is not reported in the submission, which I believe is critical.
-	The qualitative improvement shown in Fig. 3 does not echo with the quantitative one in Tab. 1. Especially, the red boxes do not help – I would say the results are mostly comparable.

**Questions:**

I am curious about how the proposed method behaves when the objects of interest contain significant holes – e.g., a donut. Currently, only the Dmask data falls into this category, and the result in Fig. 3 does not recover all the holes (which is similar to NeuS and COLMAP).

In that scenario, is the local consistency loss making sense?

---

### Note · Authors · 2024-11-16

I have read and agree with the venue's withdrawal policy on behalf of myself and my co-authors.